# Towards High Performance Chemical Vapour Deposition V_2_O_5_ Cathodes for Batteries Employing Aqueous Media

**DOI:** 10.3390/molecules25235558

**Published:** 2020-11-26

**Authors:** Dimitra Vernardou, Charalampos Drosos, Andreas Kafizas, Martyn E. Pemble, Emmanouel Koudoumas

**Affiliations:** 1Department of Electrical and Computer Engineering, School of Engineering, Hellenic Mediterranean University, 71410 Heraklion, Greece; koudoumas@hmu.gr; 2Institute of Emerging Technologies, Hellenic Mediterranean University Center, 71410 Heraklion, Greece; 3Delta Nano-Engineering Solutions, 1070 Nicosia, Cyprus; h.drosos@delta-nano.com; 4Department of Chemistry, Molecular Science Research Hub, Imperial College London, White City, London W12 0BZ, UK; a.kafizas@imperial.ac.uk; 5Grantham Institute for Climate Change and the Environment, Imperial College London, South Kensington, London SW7 2AZ, UK; 6School of Chemistry, University College Cork, T12 YN60 Cork, Ireland; martyn.pemble@tyndall.ie

**Keywords:** chemical vapour deposition, vanadium pentoxide, lithium-ion, magnesium-ion, intercalation performance

## Abstract

The need for clean and efficient energy storage has become the center of attention due to the eminent global energy crisis and growing ecological concerns. A key component in this effort is the ultra-high performance battery, which will play a major role in the energy industry. To meet the demands in portable electronic devices, electric vehicles, and large-scale energy storage systems, it is necessary to prepare advanced batteries with high safety, fast charge ratios, and discharge capabilities at a low cost. Cathode materials play a significant role in determining the performance of batteries. Among the possible electrode materials is vanadium pentoxide, which will be discussed in this review, due to its low cost and high theoretical capacity. Additionally, aqueous electrolytes, which are environmentally safe, provide an alternative approach compared to organic media for safe, cost-effective, and scalable energy storage. In this review, we will reveal the industrial potential of competitive methods to grow cathodes with excellent stability and enhanced electrochemical performance in aqueous media and lay the foundation for the large-scale production of electrode materials.

## 1. Introduction

The rapid depletion of fossil fuels and the high-level emissions of greenhouse gases in the atmosphere from the transportation are important issues of global warming [1]. This eminent global energy crisis and growing ecological concerns have led to the development of energy storage devices. These devices require a low cost, a long cycle life, excellent safety, and high energy densities with respect to unit weight and volume [2]. Rechargeable lithium-ion batteries based on organic electrolytes are widely used in portable devices due to their high energy density, cycle stability, and energy efficiency compared to other secondary batteries, such as those based on lead acid (Pb-acid), nickel-cadmium (Ni-Cd), and nickel-metal hydride (Ni-MH) [3,4,5,6]. Although such batteries have been optimized to meet the requirements of portable electronics, the intrinsic parameters of the batteries, such as cost (i.e., Co content has a significant cost and is difficult to mine properly [7,8,9]) and cycle life, in addition to their use of environmentally threatened Ni electrode materials [10], make such batteries less feasible for large scale stationary energy storage systems (EESs) [11]. Creating robust and sustainable batteries with ultra-high performance (i.e., with energy and power densities close to the theoretical values), exceptional lifetime and reliability, high safety and environmental sustainability, and large-scale production at a competitive cost remains a challenge.

Aqueous lithium-ion batteries are promising alternatives for large scale applications because they (a) require low cost since strict oxygen- and water-controlled manufacturing environments are eliminated; (b) provide a high level safety due to their non-volatility, non-toxicity, and the non-flammability of water; and (c) are capable of fast charging and high power densities due to the high ionic conductivity of the aqueous media. In particular, the ionic conductivity of the aqueous electrolytes is higher by two orders of magnitude than those of organic electrolytes, resulting in high round-trip efficiency and energy density, even with bulky and scalable electrodes [12]. Aqueous magnesium-ion batteries have also been considered as promising alternatives to lithium due to their higher abundance of magnesium (1.94%) compared to lithium resources (0.006%) [13,14]; two-electron storage per Mg-ion, offering a potential advantage in terms of volumetric capacity (3833 mAh cm^−3^ for Mg vs. 2046 mAh cm^−3^ for Li) [15]; and the analogous ionic radius of Mg^2+^ (r~0.76 Å) compared to Li^+^ (r~0.72 Å) [16,17]. Nevertheless, the diffusion of Mg^2+^ in cathode materials is slower (bivalent nature) than that of monovalent cations like Li^+^, which leads to large voltage hysteresis and a low magnesiation degree [18,19,20].

However, water has oxidation potential (oxygen evolution reaction, OER) and reduction potential (hydrogen evolution reaction, HER), which differ by a narrow voltage of 1.23 V, suppressing the operating voltage and leading to insufficient energy density [21]. Efficient ways to deal with this problem involve the passivation of water electrolysis utilizing electrodes without a catalytic effect on water splitting [22,23]. In addition, concentrated and high pH electrolytes can play an important role in the dynamics of redox reactions involving H^+^ and OH^−^. In this direction, “water-in-salt” electrolytes [24] and hydrate-melt electrolytes [25] can extend the working voltage up to 3 to 4 V, thereby overcoming water electrolysis.

Among possible electrode materials, metal oxides have attracted considerable interest due to their ability to form high energy density structures, their relatively high earth abundance, and the lack of any major safety or environmental issues associated with their use. Additionally, the high degree of the ionic character of the metal–oxygen bonds in oxides leads to a high oxidizing power of the compound and, consequently, to a high voltage battery [26]. The decrease in metal oxide particle size to the nanoscale level is assumed to result in larger electrode/electrolyte contact areas and the ability to more effectively accommodate the strain of cation intercalation/deintercalation in nanomaterials [27,28]. Vanadium pentoxide (V_2_O_5_) is one of the most promising cathode materials due to its high specific capacity, affordability, high earth abundant resources, and excellent theoretical capacity (440 mAh g^−1^) [27]. The crystal structure of V_2_O_5_ is formed by the stacking of V_2_O_5_ layers perpendicular to the c-axis via Van der Waals interactions [29]. This layered structure of V_2_O_5_ makes the intercalation of guest species into the oxide compound very feasible. However, determining how to achieve efficient electrochemical performance, including specific capacity, rate capability, and cycle life, has been very perplexing because of this structure’s low ionic diffusivity, moderate electrical conductivity [16], and narrow working potential of approximately 0–1.95 V in aqueous media [30].

In terms of manufacturing of thin film technology, chemical vapour deposition (CVD) encompasses almost all the requirements for industrialization, including electrodes for energy storage applications [31,32]. The CVD technique provides the most flexible resources for producing thin films with a high degree of control over stoichiometry, crystallinity, and uniformity, thereby permitting the utilization of precursors with low volatility [33]. This method has the advantage of being easily integrated into float-glass production lines with fast deposition rates. In addition, CVD offers excellent step coverage and excessive throughput when high aspect ratio substrates are required for ultra-large-scale integration in energy storage production.

In this review, we will summarize the cathode materials developed by CVD under atmospheric pressure along with their intercalation properties based on aqueous media with an emphasis on Li^+^ and Mg^2+^ ions. Our main objective is to illustrate the specific routes for the large-scale production of electrode materials and suggest methodologies for the development of electrodes with enhanced electrochemical performance.

## 2. Fundamental Basics on Aqueous Batteries

A battery cell consists of a cathode, an anode, an electrolyte as the media for ionic conduction, and a micro-porous polymer separator in between. The separator and the electrolyte both have negligible electronic conductance. Specifically, the separator prevents a short circuit and, at the same time, allows the diffusion of ions from the cathode to anode during the charging and the discharging processes [34]. Throughout the discharging process, the ions are extracted from the anode, pass through the separator, and are finally intercalated in the cathode. The application of a specific voltage is essential for a reversed reaction to take place during the charging state.

For storage systems based either on Li^+^ or Mg^2+^ ions, intercalation is the most commonly adopted mechanism in the development of Li and Mg batteries [35] because it maintains its structure during intercalation/deintercalation processes, assuring good cycling stability and fast ion diffusion [36,37]. The kinetics of cation intercalation are contingent on the ion mobility in the compounds, which is determined by the connectivity between sites, the sizes of the diffusion channel/cavity and ion, and the interaction strength between the ion and host structure [38,39].

### 2.1. Intercalation Cathode

An intercalation cathode is a host that can store guest ions. During the discharge process, the ions migrate from the anode to cathode generating electrons. Hence, the cathode is an important component for obtaining an efficient battery system. Studies have shown limited specific capacity and capacity degradation upon cycling due to [40,41]

H^+^ co-intercalation into the structure.Li^+^/H^+^ exchange during battery cycling.Water penetration into the structure.The slow kinetics of solid-state diffusion through the cathode material due to the high valency of Mg^2+^ ions. This is caused by the strong ionic interactions and the redistribution of divalently charged cations in the host material [42].Dissolution of active materials in the aqueous electrolytes.

To deal with these issues, several strategies have been reported, including the use of dopants, additives, modifications of the electrode/electrolyte interface (i.e., changing the electrolyte concentration or using coatings) [43,44], and nanostructured materials to decrease the diffusion length into the host cathode sites [45], thereby enlarging the surface area and providing better stress release [46,47,48,49].

#### Vanadium Pentoxide

Crystalline V_2_O_5_ consists of layers of alternating edge- and corner-sharing VO_5_ pyramids, which provide pathways for ion intercalation and deintercalation [50]. The V_2_O_5_ structure is based on the square–pyramidal coordination of V^5+^ with five oxygens and a weak V–O interaction with the sixth oxygen. These unique structural characteristics have attracted interest in energy storage applications.

In 1975, Whittingham reported reversible Li intercalation into V_2_O_5_ at room temperature [51]. A high capacity can be achieved through multilithium insertion, which, however, leads to serious structure degradation, resulting in poor cycling stability. It was found that lithium content of Li_x_V_2_O_5_ (x = 1) cannot be exceeded without reversibility degradation. Afterwards, a structural modification was found for lithium content, x > 1, where the new phase can be reversibly cycled for 0 ≤ x ≤ 2 [52]. For x ≥ 3, the V_2_O_5_ is transformed into the ω-phase with a rock-salt-type structure [52].

V_2_O_5_ has received the most attention for magnesium batteries among all layered oxides [53,54,55]. The investigation of the electrochemical intercalation of Mg^2+^ ion into orthorhombic V_2_O_5_ began in 1987 [56]. Magnesium atoms occupy the centre of the four VO_6_ octahedrons running along the α-direction of the orthorhombic lattice. Nevertheless, the capacities are much lower than expected due to spatial or redox considerations (~0.17 Mg per V_2_O_5_) [55]. The subsequent progress of V_2_O_5_ was slow because of the limitations of suitable electrolytes until a full magnesium ion battery prototype with a long cycle life was developed by Aurbach et al., 2000 [57]. After that, the use of V_2_O_5_ as a cathode material has increasingly attracted interest for post-lithium ion batteries [33,53,58].

## 3. Electrochemical Performance of V_2_O_5_ Based on Aqueous Li Electrolytes

Amorphous V_2_O_5_ was grown on a SiO_2_-precoated glass substrate at a temperature as low as 300 °C via APCVD [59]. Field emission scanning electron microscopy (FE-SEM) images indicated a flat, featureless surface (i.e., similar to the substrate) for growths at a 1.4 L min^−1^ N_2_ flow rate through the vanadium precursor bubbler, in contrast with the 0.8 L min^−1^ case, where a granular surface composed of compact crystal grains was found (Figure 1). From the deposition time and the film thickness, effective deposition rates were obtained. More precisely, the growth rate for 0.8 L min^−1^ was determined to be 4 nm min^−1^, while that for 1.4 L min^−1^ was 2.5 nm min^−1^. Therefore, above 0.8 L min^−1^, the growth rate decreased, possibly due to film decomposition or pathways reducing the precursor concentrations. In such a case, we suggest that film decomposition be applied since the film morphology became analogous with that of the substrate, as shown in Figure 1c. Moreover, no blockage occurred during the growth process.

The N_2_ flow rate was detrimental to obtaining a specific capacitance of 246 F g^−1^ since this rate significantly affected both the structure and the morphology, increasing the active area of V_2_O_5_ for an optimum value, which favoured Li ion intercalation. The specific capacitance of the as-grown coating using an N_2_ gas flow rate of 0.8 L min^−1^ was estimated by dividing the average charge intercalated by the mass of the oxide multiplied with the potential window, i.e., C = Q/ΔVm. The intercalated charge was found from the chronoamperometry measurements, as indicated in [59], while the potential window was found from the current–potential curve for an impending range of −1.000 V to +1.000 V and a scan rate of 20 mV s^−1^ in 1 M LiClO_4_/polypropylene carbonate, as shown in Figure 1b (inset). The value of 246 F g^−1^ is higher compared to that of vanadium oxide in 2 M NaCl (114 F g^−1^), 2 M LiCl (122 F g^−1^) [60], 3 M KCl (167 F g^−1^) [61], and 1 M LiClO_4_ (147 F g^−1^) [62]. However, the measured capacitance was not stable over time, showing degradation of 30% after 500 scans.

Aerosol-assisted CVD (AACVD) is another promising route for metal oxide growth. AACVD is a solution-based route that relies on the solubility of the precursor rather than its volatility in an APCVD system, thereby extending the range of the precursors that can be utilized. AACVD also permits better tailoring of the morphology, since nanoparticles and surfactants can be part of the precursor solution, which can alter the final morphology of the film [63]. From that perspective, controlling dopant incorporation is easier through the concentration of dopant in the precursor solution. Taking this advantage into consideration, we advanced with the aerosol-assisted CVD of V_2_O_5_ for different Ag loadings. In particular, vanadium bronzes of the general stoichiometry M_x_V_2_O_5_ (M = Na, Nb, Ta, K, Cu, Ag) were investigated for their capacitive performance in terms of current density, reversibility, ion storage capacity, and cyclic stability [32]. In [64], it was shown that dopant cations enhance the structural stability of V_2_O_5_ and, consequently, improve electrochemical performance.

FE-SEM was carried out to study the effect of substrate temperature on the morphology of V_2_O_5_ coatings for 5%, 10%, and 15% Ag loading at a temperature of 400 °C for a growth period of 1 h (Figure 2). Notably, it was possible to grow thin film coatings at a temperature as low as 350 °C, but with poor coverage. The as-grown samples presented rod-like structures and mostly pellet-like structures of non-uniform thicknesses and widths.

In terms of their electrochemical performance, it was not possible to obtain current–potential data because the coating was dissolved in the electrolyte even after the first Li^+^ intercalation/deintercalation scan had taken place.

For further improvement of the V_2_O_5_ electrode performance, annealing was retained for different growth periods without the presence of Ag loading. Our focus was an estimation of the critical parameters needed to obtain the required electrochemical performance. FE-SEM images of the samples grown by AACVD at 400 °C for 5 min, 10 min, 20 min, and 30 min, as well as those annealed at 600 °C for 3 h, are shown in Figure 3. In particular, the morphological characteristics indicated that times of 5 and 10 min provide particles with voids in between them, suggesting the lack of efficient vanadium oxide nucleation or adhesion. By increasing the period to 30 min, pellet-like structures were revealed (Figure 3c,d or Figure 4b), which agrees with the XRD results indicating the presence of monoclinic structures for longer periods; however, for times of 5 and 10 min, only the peaks related to the substrate were observed [65].

Figure 4a presents the cyclic voltammetry curves of the monoclinic V_2_O_5_ grown for 1 h at 400 °C and annealed at 600 °C for 3 h under a scan rate of 10 mV s^−1^ and a potential ranging between −1 V and +1 V for the 1st and the 500th scan. The curves display three oxidation and three reduction peaks, which correspond to the phase transitions between β, ε, δ, and γ [65]. These phase changes were also accompanied by colour changes of yellow → light blue → green. notably, the 1 h sample had a higher current density than the lower growth periods, which is related the the largest active material available enhancing the amount of Li-ions entering the V_2_O_5_ host lattice. Nevertheless, the current density was significantly decreased, suggesting poor stability due to non-reversible structural distortions.

Figure 4b indicates that the monoclinic β-V_2_O_5_ grown for 1 h presented similar features to the 30 min sample. For the 1 h sample, the structures lay parallel to the substrate, indicating a preferred orientation along the (200) planes (Figure 4c) [65].

Further experiments were then implemented by increasing the substrate temperature to 450 °C to enhance the coating adhesion and maintain the growth period at 1 h since the grown material at a lower substrate temperature presented better electrochemical performance compared to that grown under shorter growth periods [32]. In addition, Ag loadings were introduced to examine if the ionic conductivity would increase under the specific processing parameters. In the present study, the samples presented rod-like structures of a non-uniform thickness and width that increased in size at higher Ag loadings, achieving pellet-like structures for 5% (Figure 5) and 15% (Figure 6b). In all cases, monoclinic V_2_O_5_ was presented separately from the 5% Ag loading, showing one peak at 20° due to orthorhombic α-V_2_O_5_ [32].

Enhanced electrochemical performance was observed with Ag loading, especially for the cases of layers with high roughness. More specifically, for higher Ag content films, the amount of the incorporated charge increased along with excellent reversibility and repeatability after 500 continuous Li^+^ intercalation/deintercalation cycles. The chronopotentiometry curves of the as-grown samples at 15% Ag loading under a specific current of 400 mA g^−1^ and a potential ranging from +0.3 to −0.5 V are shown in Figure 6a. The specific discharge capacity was estimated to be 230 mAh g^−1^ with a capacity retention of 96% after 500 scans. This value is similar to that of an interlaced silver vanadium oxide-graphene hybrid [66], a one-dimensional silver vanadate nanowire hybrid with two-dimensional graphene nanosheets [67], and a peony-like silver/silver vanadate hybrid [68]. Most of these materials require rigorous mixing of toxic and flammable solvents along with the required evaporation, which adds both energy and time inputs into the electrode fabrication process. From that perspective, AACVD is advantageous since it is a route based on the deposition of materials in a more controllable manner via the thermal decomposition of precursors allowing excellent interface interaction among the materials and lowering the capital costs since the drying process is entirely circumvented.

Electrochemical impedance spectroscopy also confirmed this behaviour, showing that the transfer and diffusion of lithium ions through the cathode-electrolyte interface is easier with the highest silver content. In that case, the larger active surface area of the pellet-like structure reduced the Li^+^ intercalation rate density per unit area, which delayed the capacity loss associated with concentration polarization, resulting in a capacitance retention of 96% after 500 scans [32].

Figure 6b indicates the formation of rod-like structures with larger sizes compared to those under the 2% Ag loading. The differentiation observed among the 2% and 10% with 5% Ag loadings may be due to the co-existence of both β- and α-V_2_O_5_ phases [32].

The processing parameters utilized during the CVD routes under atmospheric pressure included the doping, the growth period, and the temperature, which were altered in a trial to enhance the electrochemical and cycling stability performance. Among these parameters, the combination of high growth temperature (450 °C) and Ag loading (15%) indicated excellent crystallinity and superior cycling stability with a specific discharge capacity of 230 mAh g^−1^, which is higher than that of nominally pure V_2_O_5_ (22.5 mAh g^−1^) [32]. This excellent performance can be credited to the pellet-like structure, which provides channels for facile Li^+^ intercalation/deintercalation along with conductivity improvements, thereby stabilizing the chemical structure during the electrochemical interactions of Li^+^. Finally, AACVD was successfully shown to be able to easily grow structures of various sizes and shapes with a high degree of control regarding uniformity over the substrate surface.

## 4. Electrochemical Performance of V_2_O_5_ Based on Aqueous Mg Electrolytes

As mentioned earlier, Li-ion batteries have been extensively studied for electronic devices due to their high energy density, cycle stability, and energy efficiency [69]. Nevertheless, their intrinsic characteristics such as cost, safety, cycle life, and even greater energy density are required to meet the increasing demands for automotive needs [70]. Multivalent ions such as Mg^2+^ are promising alternatives since they involve a two-electron transfer on V_2_O_5,_ and, when coupled with multivalent metal anodes, they can offer much higher volumetric energy density [71]. Moreover, the Mg metal anode does not form dendrites during deposition, offering safety benefits over Li metal and longer-lasting stability [72]. Additionally, Mg is more abundant than Li, which makes it cost effective.

In this work, used the advantages of the AACVD process to grow V_2_O_5_ cathode materials with excellent control over their material characteristics, including thickness, morphology, and phase, by varying the substrate temperature [33,73]. By increasing the substrate temperature from 500 to 600 °C, the coatings exhibited a compact structure with grains. However, there was differentiation in the morphologies among the samples due to improved crystallinity and the co-existence of α- and β-V_2_O_5_ at the highest growth temperature.

The electrochemical evaluation of V_2_O_5_ was studied in a three-electrode electrochemical cell using an aqueous solution of 0.075 M and MgCl_2_ as an electrolyte, sweeping the potential between −1.5 V and +1 V (Figure 7a). The stability of V_2_O_5_ was evaluated for 10,000 continuous Mg^2+^ intercalation/deintercalation scans, showing one oxidation peak at −0.61 V with the reverse reaction occurring, as indicated by the corresponding reduction peak at −1.10 V [73]. The current density was found to decrease from the first to the 2000th scan but remained constant afterwards, presenting excellent stability compared to that shown in previously published work [33]. In addition, the specific capacity of the cathode was estimated for the applied potential ranging from −1.5 V to +0.7 V for the first and the 10,000th scan (Figure 7b), employing a value of 300 mAh g^−1^ with a capacity retention of 92% after 10,000 scans and coulombic efficiency of 100%. The curves remained unchanged, indicating good stability [73]. Finally, Figure 7c presents the correlation of the power density with the energy density based on the total weight of the active material being 0.0028 g and the cell voltage and the capacity estimated for the specific current being 2.4, 3.9, and 15 A g^−1^. Here, the specific energy is comparable with that of Li-ion batteries (<300 Wh kg^−1^_cell_) and Li-S batteries (<500 Wh kg^−1^_cell_) [74].

Following these results, AACVD was shown to be a promising route to grow V_2_O_5_ coatings with strong adherence and well-defined morphological and structural characteristics for Mg-ion intercalation properties. The Li-ion intercalation properties were also tested, showing lower electrochemical performance (i.e., a lower current density by two degrees of magnitude) and stability (i.e., the electrode was dissolved in the electrolyte after the first two scans). Notably, the precursors utilized to grow the V_2_O_5_ coatings are not the same as those applied in the previous section, and, as a consequence, their characteristics and properties are different. The enhanced interaction of Mg^2+^ in V_2_O_5_ over Li^+^ in the aqueous media may be due to the high charge density of the multivalent Mg^2+^ of the Mg-ion and the slightly smaller ionic radius favouring a large number of cations diffusing in the host.

## 5. Conclusions

V_2_O_5_ is a typical intercalation compound because of the multiple valence states of vanadium and its rich structural chemistry, which enables redox-dependent properties. Taking this into consideration, we extensively investigated the intercalation properties of V_2_O_5_ for CVD processing parameters. CVD is a very resourceful method for thin film deposition since CVD properties can be simply tuned by altering the vapour composition, flow-rate, precursors, precursor concentration, pressure, substrate temperature, and growth period. Additionally, when CVD is performed under atmospheric pressure with aerosol-assistance, CVD has the advantage of requiring a relatively simple apparatus since no vacuum required and can thus be easily integrated into float-glass production lines. Additionally, AACVD is simpler to use, especially when more than one precursor has to be utilized, such as in the case of doped films.

In this review, the development of V_2_O_5_ by APCVD and AACVD under atmospheric pressure for use as a cathode for Li-ion and Mg-ion batteries was discussed, indicating the system’s capability for use in electrodes, featuring good stability and enhanced performance in aqueous media. In particular, the AACVD V_2_O_5_ coatings at 450 °C under 15% Ag loading and annealed at 600 °C for 3 h indicated superior stability with a specific discharge capacity of 230 mAh g^−1^ in the Li^+^ aqueous electrolyte, which is higher than that of nominally pure V_2_O_5_ (22.5 mAh g^−1^). In addition, the V_2_O_5_ cathode is a promising candidate for aqueous Mg^2+^ batteries, offering excellent stability up to 10,000 scans.

Nevertheless, there is space for conductivity and stability improvement. In our laboratory, we are currently examining other prospects. Using graphene as a precoating is a good option since graphene can act as a cutout to build a porous network, thereby improving the electrical properties and charge transfer pathways of oxides. A major advantage of graphene is the presence of oxygen-containing groups on its edges and surfaces, which strongly influence the size, shape, and distribution of the metal oxide nanostructures on the graphene. Titanium-based compounds as precoating are another route able to combine with V_2_O_5_ since these compounds exhibit the advantages of a small volume expansion ratio upon intercalation/deintercalation of the Li ions and respectable cyclic stability.

However, there is still remaining materials chemistry to consider for the development of cathode materials with excellent intercalation properties. Hence, it is important to realize the potential of CVD systems under atmospheric pressure for the realization of large-scale full battery cells at a low cost. It is important to note that this overall cost reduction can only be achieved if the costs of each component are also reduced.

## Figures and Tables

**Figure 1 molecules-25-05558-f001:**
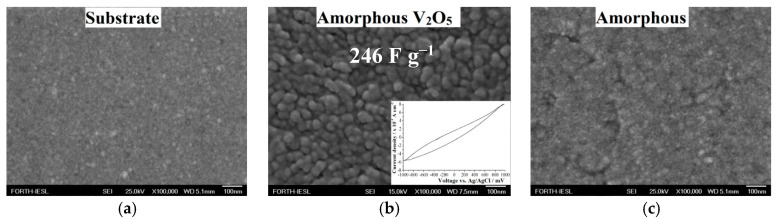
FE-SEM images of vanadium oxides grown by Atmospheric Pressure Chemical Vapour Deposition at 300 °C on (**a**) a SiO_2_-precoated glass substrate for (**b**) 0.8 L min^−1^ and (**c**) 1.4 L min^−1^ N_2_ flow rates through the vanadium precursor bubbler. Inset (**b**) The cyclic voltammetry curve of the as-grown APCVD vanadium oxide for the 0.8 L min^−1^ N_2_ flow rate for a potential ranging of −1.000 V to +1.000 V and a scan rate of 20 mV s^−1^ in 1 M LiClO_4_/polypropylene carbonate.

**Figure 2 molecules-25-05558-f002:**
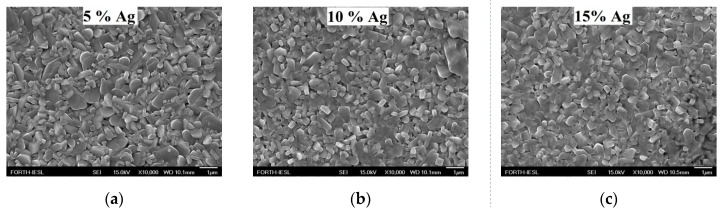
FE-SEM images of V_2_O_5_ grown by AACVD under atmospheric pressure for (**a**) 5%, (**b**) 10%, and (**c**) 15% Ag loading at 400 °C for 1 h.

**Figure 3 molecules-25-05558-f003:**
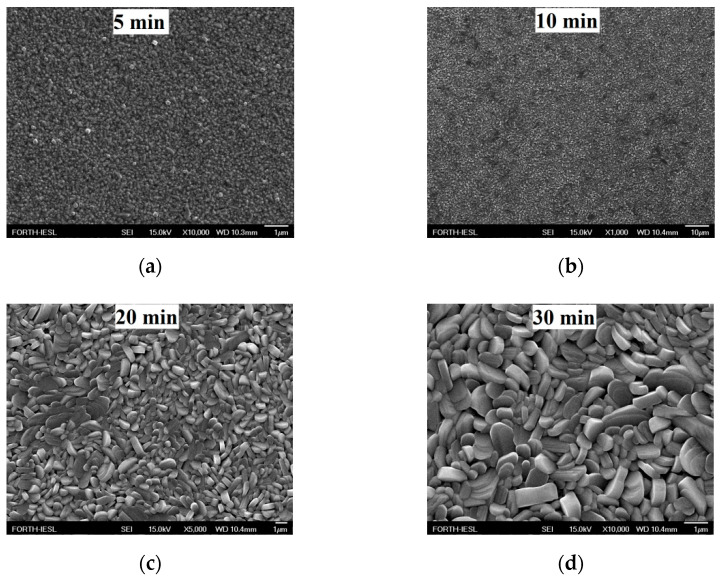
FE-SEM images of V_2_O_5_ grown by AACVD at 400 °C for growth periods of (**a**) 5 min, (**b**) 10 min, (**c**) 20 min, and (**d**) 30 min and annealed at 600 °C for 3 h.

**Figure 4 molecules-25-05558-f004:**
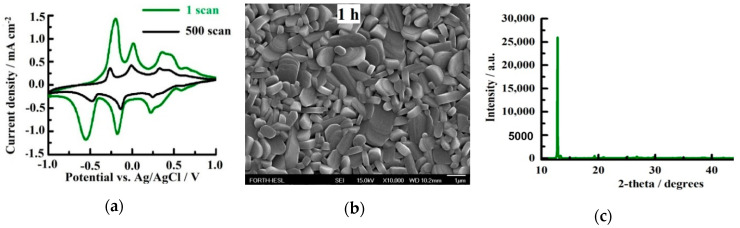
(**a**) Current-potential curves of the V_2_O_5_ grown by AACVD at 400 °C for a growth period of 1 h and annealed at 600 °C for 3 h and (**b**) the FE-SEM images of the sample; (**c**) XRD pattern of the sample.

**Figure 5 molecules-25-05558-f005:**
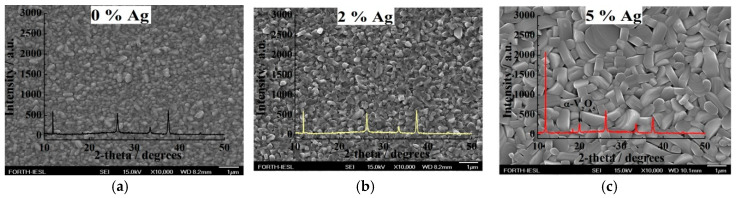
XRD patterns overlapped over FE-SEM images of the V_2_O_5_ grown at 450 °C for 1 h via AACVD under atmospheric pressure for (**a**) 0% (**b**) 2%, and (**c**) 5% Ag loading and annealed at 600 °C for 3 h.

**Figure 6 molecules-25-05558-f006:**
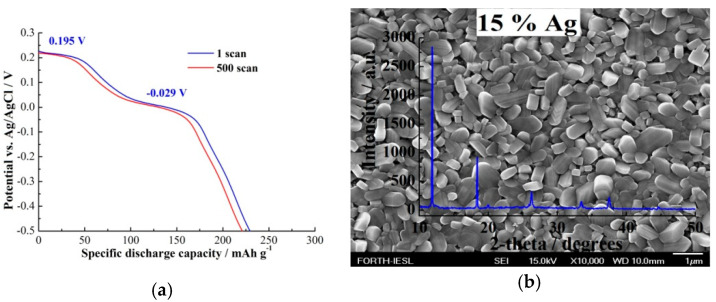
(**a**) The chronopotentiometry curves of the V_2_O_5_ samples at 450 °C via AACVD at atmospheric pressure for 15% Ag loading and annealed at 600 °C for 3 h under a specific current of 400 mA g^−1^ and potential ranging from +0.3 to −0.5 V for the 1st and the 500th scan. (**b**) XRD overlapped over the FE-SEM image of the sample.

**Figure 7 molecules-25-05558-f007:**
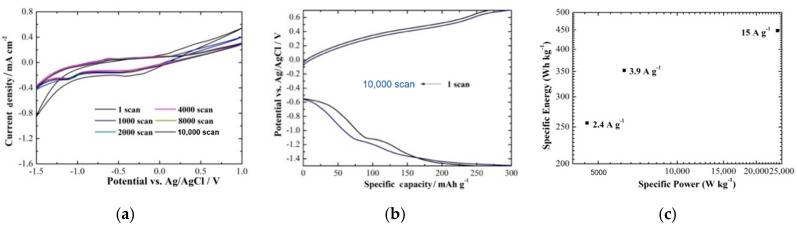
(**a**) Current–potential curves of the V_2_O_5_ coatings grown by AACVD at 600 °C sweeping the potential between −1.5 V and +1.0 V in 0.075 M MgCl_2_ for 1, 1000, 2000, 4000, 8000, and 10,000 scans. (**b**) Specific capacity of the cathodes for a potential ranging −1.5 V to +0.7 V for 1 and 10,000 scans. (**c**) Ragone plot for a specific current of 2.4, 3.9, and 15 A g^−1^.

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
