# Peer review of "Towards High Performance Chemical Vapour Deposition V2O5 Cathodes for Batteries Employing Aqueous Media"

_molecules, 2020, doi:10.3390/molecules25235558_

Round 1

Reviewer 1 Report

The manuscript entitled "Towards high performance CVD V2O5 cathodes for batteries employing aqueous media" which was presented to me is showing the structure characteristic and electrochemical performance of V2O5. The manuscript is publishable if authors can answer the following questions that I give, which might improve the final draft.

  1. The author explained that the difference in crystallinity of V2O5 by various experimental conditions exists, and that electrochemical performance differences exist depending on it. However, the absence of XRD data makes it impossible to assert credibility in the author's opinion. It would be nice to talk about electrochemical performance with XRD data from V2O5.
  2. Are there any additional electrochemical analysis results that can be found in the aqueous Mg electrodes? The data is too insufficient.

Author Response

Dear Reviewer 1,

Thank you very much for your response and helpful suggestions regarding the manuscript entitled:

“Towards high performance CVD V2O5 cathodes for cathodes employing aqueous media”

we would like to publish in Molecules. We have done revisions in our original manuscript and we are resubmitting our work hoping that we have fully complied with your recommendations. The highlighted revision encloses all changes in bold and underline.

Reviewer’s comments:

The manuscript entitled "Towards high performance CVD V2O5 cathodes for batteries employing aqueous media" which was presented to me is showing the structure characteristic and electrochemical performance of V2O5. The manuscript is publishable if authors can answer the following questions that I give, which might improve the final draft.

  1. The author explained that the difference in crystallinity of V2O5 by various experimental conditions exists, and that electrochemical performance differences exist depending on it. However, the absence of XRD data makes it impossible to assert credibility in the author's opinion. It would be nice to talk about electrochemical performance with XRD data from V2O5.

Response: Thank you very much for the comment. We did not include XRD patterns because they are presented in the papers cited in our work. In the revised manuscript, we added some XRD data along with the corresponding discussion.

  1. Are there any additional electrochemical analysis results that can be found in the aqueous Mg electrodes? The data is too insufficient.

Response: Thank you very much for the comment. We agree with the reviewer that more discussion is required in the section related with the Mg electrodes. Hence, we have added the necessary information in the revised manuscript.

With our best regards

Reviewer 2 Report

The authors present an interesting and well-written review of novel Vanadium-based cathodes with environmental benign materials. Informative and interesting to read. Though, in my opinion some respective point should be revised and references updated in order to increase the impact of this work

  1. Please do not use abbreviation in your title.

  1. Organic rechargeable lithium-ion batteries are widely used in 38 portable devices due to their highest energy density, cycle stability and energy efficiency compared 39 to other secondary batteries such as those based on lead acid (Pb-acid), nickel-cadmium (Ni-Cd) and 40 nickel-metal hydride (Ni-MH)”

                            The term “organic” is not conventional for state of the art batteries, i.e. Li ion batteries. Organic compounds are the electrolyte solvents, while the residual components like the Li salt and electrodes are inorganic. The term organic is used for next generation materials, e.g. cathodes based on organic compounds, which, though, are at research state and not state of the art. Please revise using following references:

                                Doi 10.1021/Cr020731c

                             Doi 10.1021/Cr030203g

Doi 10.1021/cm901452z

  1. Aqueous lithium-ion batteries are promising alternatives for large scale applications because they (a) require low cost since strict oxygen- and water-controlled manufacturing environments are eliminated (b) provide high safety due to the non-volatility, non-toxicity and non-flammability of water and (c) are capable of fast charging and high power densities due to the high ionic conductivity of aqueous media. How significant is the difference of Co and Ni with respect to the environmental issues and costs? I recommend to specify this sentence and to include suitable references.”

Li ion batteries imply a rocking chair chemistry overwhelmingly with a graphite-based anode. This anode reaches potentials of almost 0 V vs Li/Li+, i.e. maximal reductive. How can be water suitable at these conditions without releasing hazardous gases like hydrogen. To be honest, for me this is counterintuitive, at least I am not up to date with this relation. So please provide respective literature for readers like me to get updated. At least regarding the theoretical aspects, please consider following reference for this passage:

https://doi.org/10.1007/s41061-018-0196-1

  1. and the analogous ionic radius of Mg2+ (r~0.89 Å) as compared with Li+ (r~0.92 Å).”

Ionic radius of Li+ is 0.76 Å according the literature of Li ion batteries for example in following reference:

doi:10.1002/ente.201700068

  1. This layered structure of V2O5 makes the intercalation of guest species into the oxide compound quite feasible. However, in order to achieve efficient electrochemical performance, including specific capacity, rate capability and cycle life, has been very perplexing because of its low ionic diffusivity and moderate electrical conductivity [16].”

What about the voltage, as an other crucial factor for energy density. Please provide the missing information with appropriate reference.

  1. A battery cell consists of a cathode, an anode, an electrolyte and a micro-porous polymer separator in between, to prevent short circuit and, at the same time, allow the diffusion of ions from cathode to anode during the charging and the discharging process.”

A separator is a material which prevents contact of the electrodes while the “electrolyte” is the media for ionic conduction, both with negligible electronic conductance. A separator without electrolyte would not work. Separator and electrolyte at the same time are solid electrolytes. Please revise accordingly using following reference

https://doi.org/10.1002/adfm.202006289

  1. “Figure 4”

The term “potential“ in the x-axis implies a reference electrode (three electrode cell setup), for example Li electrode. In this case this should be terminated e.g. “potential vs Li/Li+ / V”. However, when no reference electrode is used, i.e. in a two electrode setup, than please use “voltage / V” and revise accordingly also for residual figures. These relations can be found in https://doi.org/10.1016/j.mattod.2019.07.002

Author Response

Dear Reviewer 2,

Thank you very much for your response and helpful suggestions regarding the manuscript entitled:

“Towards high performance CVD V2O5 cathodes for cathodes employing aqueous media”

we wish to publish in Molecules. We have done revisions in our original manuscript and we are resubmitting our work hoping that we have fully complied with your recommendations. The highlighted revision encloses all changes in bold and underline.

Reviewer’s comments:

The authors present an interesting and well-written review of novel Vanadium-based cathodes with environmental benign materials. Informative and interesting to read. Though, in my opinion some respective point should be revised and references updated in order to increase the impact of this work

  1. Please do not use abbreviation in your title.

Response: Thank you for the comment. The title has been changed accordingly.

  1. “Organic rechargeable lithium-ion batteries are widely used in 38 portable devices due to their highest energy density, cycle stability and energy efficiency compared 39 to other secondary batteries such as those based on lead acid (Pb-acid), nickel-cadmium (Ni-Cd) and 40 nickel-metal hydride (Ni-MH)”

The term “organic” is not conventional for state of the art batteries, i.e. Li ion batteries. Organic compounds are the electrolyte solvents, while the residual components like the Li salt and electrodes are inorganic. The term organic is used for next generation materials, e.g. cathodes based on organic compounds, which, though, are at research state and not state of the art. Please revise using following references: Doi 10.1021/Cr020731c, Doi 10.1021/Cr030203g, Doi 10.1021/cm901452z

Response: Thank you for the comment. We have proceeded with the necessary changes in Introduction and we have included the suggested references in the revised manuscript.

  1. Aqueous lithium-ion batteries are promising alternatives for large scale applications because they (a) require low cost since strict oxygen- and water-controlled manufacturing environments are eliminated (b) provide high safety due to the non-volatility, non-toxicity and non-flammability of water and (c) are capable of fast charging and high power densities due to the high ionic conductivity of aqueous media.

How significant is the difference of Co and Ni with respect to the environmental issues and costs? I recommend to specify this sentence and to include suitable references.”

Response: Thank you for the comment. We have proceeded with the necessary changes in Introduction and we have included the related references in the revised manuscript.

Li ion batteries imply a rocking chair chemistry overwhelmingly with a graphite-based anode. This anode reaches potentials of almost 0 V vs Li/Li+, i.e. maximal reductive. How can be water suitable at these conditions without releasing hazardous gases like hydrogen. To be honest, for me this is counterintuitive; at least I am not up to date with this relation. So please provide respective literature for readers like me to get updated. At least regarding the theoretical aspects, please consider following reference for this passage:

https://doi.org/10.1007/s41061-018-0196-1

Response: Thank you for the comment. We have proceeded with the necessary changes in Introduction and we have included the related references in the revised manuscript.

  1. and the analogous ionic radius of Mg2+ (r~0.89 Å) as compared with Li+ (r~0.92 Å).”

Ionic radius of Li+ is 0.76 Å according the literature of Li ion batteries for example in following reference:

doi:10.1002/ente.201700068

Response: Thank you for the comment. We have found that the ionic radius of Mg2+ is 0.76 Å and Li+ is 0.72 Å according with the related references (10.1002/adma.201806510, 10.1002/smtd.201800272) that have been included in the revised manuscript.

  1. “This layered structure of V2O5 makes the intercalation of guest species into the oxide compound quite feasible. However, in order to achieve efficient electrochemical performance, including specific capacity, rate capability and cycle life, has been very perplexing because of its low ionic diffusivity and moderate electrical conductivity [16].”

What about the voltage, as an other crucial factor for energy density. Please provide the missing information with appropriate reference.

Response: Thank you for the comment. We have done the corresponding change in Introduction along with the appropriate reference in the revised manuscript.

  1. A battery cell consists of a cathode, an anode, an electrolyte and a micro-porous polymer separator in between, to prevent short circuit and, at the same time, allow the diffusion of ions from cathode to anode during the charging and the discharging process.”

A separator is a material which prevents contact of the electrodes while the “electrolyte” is the media for ionic conduction, both with negligible electronic conductance. A separator without electrolyte would not work. Separator and electrolyte at the same time are solid electrolytes. Please revise accordingly using following reference

https://doi.org/10.1002/adfm.202006289

Response: Thank you for the comment. We have done the corresponding change in the revised manuscript.

7. “Figure 4” The term “potential” in the x-axis implies a reference electrode (three electrode cell setup), for example Li electrode. In this case this should be terminated e.g. “potential vs Li/Li+ / V”. However, when no reference electrode is used, i.e. in a two electrode setup, than please use “voltage / V” and revise accordingly also for residual figures. These relations can be found in https://doi.org/10.1016/j.mattod.2019.07.002

Response: Thank you for the comment. We have done the corresponding changes in Figures in the revised manuscript.

With our best regards

Round 2

Reviewer 1 Report

The points mentioned in the first revision were well-corrected.

Therefore, there is no additional modification.

Reviewer 2 Report

The authors provided good work and revised appropriately. The manuscript can be accepted in present form.